# ABO Blood Groups and the Incidence of Complications in COVID-19 Patients: A Population-Based Prospective Cohort Study

**DOI:** 10.3390/ijerph181910039

**Published:** 2021-09-24

**Authors:** Salvador Domènech-Montoliu, Joan Puig-Barberà, Maria Rosario Pac-Sa, Paula Vidal-Utrillas, Marta Latorre-Poveda, Alba Del Rio-González, Sara Ferrando-Rubert, Gema Ferrer-Abad, Manuel Sánchez-Urbano, Laura Aparisi-Esteve, Gema Badenes-Marques, Belén Cervera-Ferrer, Ursula Clerig-Arnau, Claudia Dols-Bernad, Maria Fontal-Carcel, Lorna Gomez-Lanas, David Jovani-Sales, Maria Carmen León-Domingo, Maria Dolores Llopico-Vilanova, Mercedes Moros-Blasco, Cristina Notari-Rodríguez, Raquel Ruíz-Puig, Sonia Valls-López, Alberto Arnedo-Pena

**Affiliations:** 1Emergency Service Hospital de la Plana, Health Department 3, 12540 Vila-Real, Spain; pttcarmen@hotmail.com (S.D.-M.); martalapo@hotmail.com (M.L.-P.); manu.msu@gmail.com (M.S.-U.); gemabamar@hotmail.com (G.B.-M.); belencerveraferrer@hotmail.es (B.C.-F.); ursuclerig@gmail.com (U.C.-A.); lornagl78@gmail.com (L.G.-L.); jovasal1987@gmail.com (D.J.-S.); llopivila@hotmail.com (M.D.L.-V.); raquelruizpuig@gmail.com (R.R.-P.); Sonia.valls.lopez@gmail.com (S.V.-L.); 2Vaccines Research Area FISABIO, 46020 Valencia, Spain; jpuigb55@gmail.com; 3Public Health Center, Health Department 2, 12003 Castello de la Plana, Spain; charopac@gmail.com; 4Health Centers I and II, Health Department 2, 12530 Borriana, Spain; vidalutrillaspaula@gmail.com (P.V.-U.); delrio_alb@gva.es (A.D.R.-G.); sfr1812@gmail.com (S.F.-R.); gferrer@uji.es (G.F.-A.); 5Carinyena Health Center, 12540 Vila-Real, Spain; lauraaparisiesteve@gmail.com (L.A.-E.); mercedesmb1094@gmail.com (M.M.-B.); 6Health Center, Health Department 3, 12200 Onda, Spain; claudiadb1294@hotmail.com (C.D.-B.); notari_cri@gva.es (C.N.-R.); 7Health Center, Health Department 3, 12600 La Vall d’Uixó, Spain; fontalcarcel93@hotmail.com; 8Villa Fátima School, Health Department 3, 12530 Borriana, Spain; carmendole@hotmail.com; 9Department of Health Science, Public University Navarra, 31008 Pamplona, Spain; 10Epidemiology and Public Health (CIBERESP), 28029 Madrid, Spain

**Keywords:** COVID-19, post-COVID-19, complications, symptoms, ABO blood groups, incidence, cohort, population based

## Abstract

After a COVID-19 outbreak in the Falles festival of Borriana (Spain) during March 2020, a cohort of patients were followed until October 2020 to estimate complications post-COVID-19, considering ABO blood groups (ABO). From 536 laboratory-confirmed cases, 483 completed the study (90.1%) carried by the Public Health Center of Castelló and the Emergency and Microbiology and Clinical Analysis of Hospital de la Plana Vila-real. The study included ABO determination and telephone interviews of patients. The participants had a mean age of 37.2 ± 17.1 years, 300 females (62.1%). ABO were O (41.4%), A (45.5%), B (9.1%), and AB (3.9%). We found no difference in the incidence of COVID-19 infections. A total of 159 (32.9%) patients reported one or more post-COVID-19 complications with divergent incidences after adjustment: O (32.3%), A (32.6%), B (54.1%), and AB (27.6%); B groups had more complications post-COVID-19 when compared with O group (adjusted relative risk [aRR] 95% confidence interval [CI] 1.68, 95% CI 1.24–2.27), and symptoms of fatigue (1.79, 95% CI 1.08–2.95), myalgia (2.06, 95% CI 1.10–3.84), headache (2.61, 95% CI 1.58–4.31), and disorder of vision (4.26 95% CI 1.33–13.60). In conclusion, we observed significant differences in post-COVID-19 complications by ABO, with a higher incidence in B group. Additional research is justified to confirm our results.

## 1. Introduction

Ongoing symptomatology in COVID-19 patients after the acute phase of the illness is frequent. It has been characterized as a syndrome of long or persistent COVID, affecting both children and adults [1,2,3]. The research on the risk and protective factors of infection and severity of COVID-19 is crucial in the second year of the pandemic. Thus far, many potential factors have been considered [4,5].

Among these factors, the ABO blood groups (ABO) are being studied with intensity [6,7,8,9], and it has been recommended to determine the ABO of COVID-19 patients to improve medical care [10].

There are, however, conflicting results about the role of ABO and the risk of COVID-19 disease. Some studies found the A and B blood groups related to a higher risk of infection, complications, and mortality [11,12,13,14,15]. In contrast, other studies have not found differences in disease severity among the O and A groups [16], whereas other studies have found the O group as protective, compared to the non-O blood groups. Finally, large and well-designed studies have not found that the ABO blood groups are a risk factor for COVID-19 infection or severity [17,18].

Several conditions could explain this situation of conflicting results, including diversity of sample size, effect size, multiple confounders, publication bias, or even chance distributions [17,19,20]. Complications considering ABO and the follow-up of patients with a mild illness have been less studied [21,22,23,24].

This study aimed to estimate the frequency of persistent symptoms and complications in COVID-19 patients and its associations with ABO blood groups.

## 2. Materials and Methods

After a COVID-19 outbreak in the Falles festival with several mass gathering events (MGEs) in Borriana (Spain) during March 2020 [25], we followed a population-based prospective cohort of patients until October 2020 [26] to estimate disease evolution and incidence of complications post-COVID-19, and its relationship with participants ABO blood group. The study was carried by the Public Health Center of Castelló, and the Emergency and Microbiology and Clinical Analysis Services of Hospital de la Plana Vila-real (Spain).

We performed the first study of the COVID-19 outbreak during May–June 2020 through a serological survey and questionnaire interview [25]; we included 1338 people in the study, and we found 570 COVID-19 patients with 536 patients, laboratory-confirmed tests: electrochemiluminescence immunoassay (ECLIA) (Elecsys^®^, Mannheim, Germany, Anti-SARS-CoV-2, Roche Diagnostics) [27], in 514 patients, lateral flow immunochromatographic assay (LFIC), in 15 patients, and reverse transcriptase–polymerase chain reaction (RT-PCR) in 39 patients [25]. ABO was determined by the gel test (ID-Card ABO/RhD, DiaMed GmbH, Bio-Rad Laboratories Switzerland) [28].

In October 2020, health staff of the Hospital de la Plana Vila-real, and health centers of Borriana, Vila-real, Onda, and La Vall d’Uixo conducted a telephone interview of each participant to obtain information about their health situation, medical assistance, illness’s evolution, symptoms post-COVID-19, and duration. In addition, we acquired data from the May–June questionnaire on age, sex, weight, height, body mass index (BMI) (kg/m^2^), occupation, level of physical exercise, smoking habits, consumption of alcohol, chronic illness, and COVID-19 exposures. COVID-19 exposition included the following features: see a person with a cough at MGEs, attendance MGEs ≥ 2, contact with a COVID-19 case, and family with COVID-19 case.

### Statistical Methods

The expected incidence of COVID-19 infection by the ABO was estimated from the distribution of ABO in three sources: two blood donor studies of Catalonia and Navarra [13,29] and one study of an active general population of the Spanish Mediterranean zone [30]. We compared the observed ABO distribution with the expected distribution through Fisher’s exact test. Distributions of ABO from the three sources were applied to the 1338 participants in the COVID-19 outbreak study [25] to obtain an incidence rate by each ABO. We used the O group as the reference to estimate relative risks (RR) and 95% confidence interval (CI) by Poisson regression.

Reported complications and symptoms were the dependent variables, and ABO was the predictive variable. We used the Chi2, Fisher’s exact test, Wilcoxon matched-pairs signed-ranks test, and Kruskal–Wallis tests for unadjusted comparisons. After a review of medical literature, we used directed acyclic graphs [31], and identified age, sex, lifestyle, and COVID-19 exposures as confounding factors [32], and adjusted our models accordingly using inverse probability weight regression [33]. We performed all calculations with the statistical program STATA^®^ version 14.

Following up this cohort was part of the public health surveillance as a prolongation of the COVID-19 outbreak control measures [25] and the response of the COVID-19 pandemic. It was exempted from Ethics Review Board approval’s protocol according to the Spanish legislation. The study was approved by the director of the Public Health Center of Castelló and the management of the Health Department of La Plana. All participants or the parents of the minors provided their informed written consent to be included in the study.

## 3. Results

The participation rate was 90.1% (483/536) considering the patients with laboratory-confirmed COVID-19 in the COVID-19 outbreak. The mean age was 37.2 ± 17.1 years (rank 1–81), with 300 females (62.1%) and 183 males (37.9%). ABO blood groups distribution was O (41.4%), A (45.5%), B (9.1%), and AB (3.9%). The subjects’ characteristics by ABO are shown in Table 1. We found no appreciable differences among ABO by demographic, lifestyle, and COVID-19 exposure.

We found no difference in the incidence of SARS-CoV-2 when comparing the observed proportion of ABO and the expected proportion from blood donors of Catalonia (*p* = 0.247) and Navarra (*p* = 0.089), and an active general population (*p* = 0.366) (Table 2). However, RRs of no-O groups presented a higher incidence than the estimated O group with a range of 1.06–2.06 and was significant when applying the two blood donor sources to the participants in the COVID-19 outbreak but not when using the active general population.

Symptoms and duration of the COVID-19 illness among ABO are shown in Table 3. B group experienced a higher proportion of symptomatic disease, medical consultation, hospitalization, and a longer duration of illness. B group reported a higher presence of symptoms such as diarrhea, vomits, weakness, headache, and myalgia. Fever and loss of smell/taste presented significant differences with the other blood groups, (*p* = 0.020) and (*p* = 0.016), respectively. There was no difference in habitual health status before and after the COVID-19 considering the ABO blood group. We found significant differences in the O and A groups between the before and after the illness in health status (*p* = 0.000) and (*p* = 0.001); in contrast, little to no difference for the B group (*p* = 0.180) and AB group (*p* = 0.414).

A 32.9% of patients (159/483) reported at least one complication (Table 4) with a mean duration of 160.9 ± 45.6 days. The incidence of complications was higher in the B group (50%), compared to the other ABO (31.2%) with a marginally non-significant difference (*p* = 0.072). In October 2020, 81.8% of patients (395/483) had recovered to their former health, but recovery was less frequent in the B group (70.5%), compared to the other ABO (82.9%) (*p* = 0.047). In addition, 83.2% (402/483) reported that their health status was the same as before the illness; this was also lower in subjects of the B group (70.5%) when compared to the subjects with other ABO (84.3%) (*p* = 0.025).

In October 2020, 53.4% of patients reported at least one symptom (258/483) without difference among ABO (*p* = 0.586) (Table 4). Reported symptoms of fatigue, abdominal pain, muscle pain, loss of smell/taste, headache, the difficulty to solve simple math operations, and skin lesions was higher in the B group. The AB group had a lower incidence of complications and reported symptoms than the B group, and compared with the O group, had high recovery and return health status as before.

We show adjusted incidence rate (aIR) and adjusted RR (aRR) of complications and reported symptoms in Table 5. The B group presented higher aIR than the other ABO in complications, lower recovery, and return health status as before with aRR of 1.68 (95% CI 1.24–2.27), 0.86 (0.70–1.03), and 0.85 (0.71–1.04).

Compared with the O-group subjects, the B-group subjects reported more frequently symptoms of fatigue (29.0% versus 16.2% aRR = 1.79 95% CI 1.08–2.95), muscle pain (21.1% versus 9.8% aRR = 2.06 95% CI 1.10–3.84), headache (36.4% versus 13.9% aRR = 2.61 95% CI 1.58–4.31), disorder vision (12.8% versus 3.0% aRR = 4.26, 95% CI 1.33–13.60), and medical consultation in the acute illness (56.7% versus 42.5% aRR = 1.33 95% CI 1.01–1.75). Loss of the smell/taste was higher in the B group but with a marginally non-significant difference (26.0% versus 14.8% aRR = 1.75, 95% CI 0.95–3.23). Brain fog, as a summary of mental symptoms (Table 5), was higher in the B group but not significant (18.1% versus 12.3% aRR = 1.46 95% CI 0.75–2.73). On the other hand, the AB group’s small size prevented obtaining valid estimations.

## 4. Discussion

Our results suggest that ABO had not an appreciable impact on the incidence of COVID-19 infection, with the B-group subjects experiencing a higher incidence of complications and reported symptoms than the O group at six months after the acute COVID-19 illness.

Regarding the incidence of SARS-CoV-2 infection in ABO, our results are in line with several studies where no effect of ABO could be found with suitable controls for comparison [17,18,34,35]. The protection of the O group among ABO, which could be found in several studies [29,36,37], is overcome following the Ellis model of ABO incompatibility and SARS-CoV-2 transmission [38]; in the first stages of the epidemic, the non-O groups are more infected, but after, when the majority of the population is infected, the O group has a similar infection rate.

Most ABO studies have been focused on the incidence and severity of COVID-19 [17], and few studies on symptoms and complications post-COVID-19. Some studies found that the O group offered protection against COVID-19 infection, whereas A-group subjects experienced a higher risk of adverse outcomes [12,13,39,40], with more cardiovascular complications in the A-group subjects [41], whereas subjects in the B group experienced more severe illnesses and higher fatality rates [11,15,22,41,42], but no ABO blood group differences in mortality were found in other studies [43,44,45].

In our study, the B-group subjects reported more symptoms and severity of the disease than the other groups. This pattern has been associated with post-COVID-19 complications [46,47]. Complications and reported symptoms here are concordant with the post-COVID-19 disease despite a mild illness in most patients. Hair loss, fatigue, smell/taste loss, headache, muscle pain, insomnia, and anxiety symptoms agree with the post-COVID-19 disease definition [48,49]. In general, physical symptoms (fatigue, muscle pain, headache, lost smell/taste, and vision disorder) had a higher incidence in the B group than mental symptoms (anxiety, depression, insomnia) when compared with other ABO. No difference in complications and reported symptoms were found between O and A groups, in contrast with other studies [23,50,51,52].

Le Pendu et al. [53], in a review of the association between ABO and COVID-19, concluded that the O group provided some protection in comparison with non-O groups, and this could be mediated either by natural anti-A and anti-B antibodies or by a lower efficiency of furin cleavage in the O group [54,55]. In addition, Bloch et al. [56] found that blood donors of group B had higher neutralizing antibody titers than the other ABO. According to this study [56], a possibility is cross reactivity of the virus and the B antigen, and this increases antibody production, but the cause is unknown. In our previous study [26], the O group presented lower persistence of anti-SARS-CoV-2 antibodies than the non-O groups, but we found no difference in antibody levels between ABO blood groups.

Concerning infectious diseases, the O group could have special protection derived from anti-A/anti-B actions of humoral innate immunity [57,58]. In addition, the O group had some physiological advantages such as endurance running, compared with non-O-groups [59]. ABO differences in respect to health and disease are not entirely understood. Still, some studies found that O group subjects have lower thrombosis and more hypertension disease than no-O groups, and the A group has more risk of cardiovascular disease [60,61,62].

The study had some important aspects, including a population-based prospective cohort design, high participation rate, a follow-up of 6 months, and control of potential confounding factors, considering that the exposure to the virus and the duration time could play a crucial role in the clinical course of the disease [63].

The study has some limitations; measured manifestations were reported by questionnaire, so we could not discard information, recall bias, and residual confounding. COVID-19 is a new disease, and some aspects could not be considered in advance. Most patients had a mild illness, and they may not be representative of the COVID-19 patients. A small sample of the AB group prevented definitive estimations in this blood group. The study was focused on COVID-19 patients, and we could not compare our findings with the COVID-19 negative member of the cohort.

Our results support the active medical follow-up of COVID-19 patients considering the high level of symptoms persistence [64]. In our study, the B group appears associated with a higher risk of prolonged symptoms, whereas the O group subjects experienced lower affectation. Considering that there are few studies on ABO blood groups and COVID-19 complications, further research is justified to improve our understanding of the ABO relationship with COVID-19 [65,66,67].

## 5. Conclusions

ABO blood group patients presented significant differences in post-COVID-19 complications with a more severe course observed in the B group. Additional research is justified to confirm our results.

## Figures and Tables

**Table 1 ijerph-18-10039-t001:** Distribution of characteristics ABO blood groups in the subjects included October 2020. Borriana COVID-19 cohort 2020.

Variables	ABO Blood Groups *n* = 483				*p*-Value
	O *N* = 200 *N* (%)	A *N* = 220 *N* (%)	B *N* = 44 *N* (%)	AB *N* = 19 *N* (%)	
Female Sex	127 (63.5)	131 (59.5)	31 (70.5)	11 (57.9)	0.528
Age (years) mean ± standard desviation	37.3 ± 15.5	37.4 ± 16.9	37.9 ± 18.0	33.7 ± 13.5	0.528
0–14	23 (11.5)	24 (10.9)	5 (11.4)	1 (5.3)	
15–24	39 (19.5)	36 (16.4)	9 (20.5)	5 (26.3)	
25–34	21 (10.5)	34 (15.5)	3 (6.8)	2 (10.5)	
35–44	40 (20.0)	39 (17.7)	9 (20.5)	9 (47.4)	
45–54	40 (20.0)	53 (24.1)	11 (25.0)	2 (10.5)	
55–64	28 (14.0)	27 (12.3)	6 (13.6)	0 (0)	
65 and over	9 (4.5)	7 (3.2)	1 (2.3)	0 (0)	
Occupation ^1,2^					
Occupation I-II	60 (30.3)	63 (28.6)	17 (39.5)	4 (21.1)	0.444
Occupation III-VI	138 (69.7)	157 (71.4)	26 (60.5)	15 (78.9)	
Physical exercise	115 (57.5)	135 (61.4)	29 (67.4)	10 (52.6)	0.656
Alcohol consumption ^3^	44 (23.0)	45 (20.8)	12 (27.9)	7 (36.8)	0.336
Smoking ^4^					
No smoking	119 (62.6)	135 (62.8)	26 (60.5)	16 (84.2)	0.567
Ex smoking	42 (22.1)	52 (24.2)	11 (25.6)	1 (5.3)	
Current smoker	29 (15.3)	28 (13.0)	6 (13.9)	2 (10.5)	
Body Mass Index (Kg/m^2^) ^5^					
<18.5	20 (10.1)	15 (6.9)	5 (11.6)	1 (5.3)	0.321
18.5–24.9	92 (46.2)	86 (39.6)	20 (46.5)	11 (57.9)	
25.0–29.9	51 (25.6)	80 (36.9)	11 (25.6)	6 (31.6)	
≥30.0	36 (18.1)	36 (16.6)	7 (16.3)	1 (5.3)	
Chronic illness ^6^	66 (33.2)	81 (37.3)	14 (31.8)	5 (26.3)	0.698
Exposure COVID-19					
See a person with a cough at mass gathering events ^7^	91 (45.7)	93 (43.1)	15 (34.9)	6 (31.6)	0.442
Attendance mass gathering events ≥ 2	121 (60.5)	135 (61.4)	27 (61.4)	11 (57.9)	0.989
Contact COVID-19 case ^8^	161 (81.7)	179 (82.5)	35 (81.4)	14 (73.7)	0.774
Family COVID-19 case ^9^	127 (63.5)	131 (59.5)	32 (74.4)	13 (68.4)	0.292

^1^ Missing information 3 participants; ^2^ occupation groups I-II: professional, managerial, and technical occupations; groups III-VI: skilled, non-manual or manual, partly skilled, unskilled occupations; ^3^ missing information 14 participants; ^4^ missing information 16 participants; ^5^ missing information 5 participants; ^6^ missing information 4 participants; ^7^ missing information 6 participants; ^8^ missing information 7 participants; ^9^ missing information 1 participant.

**Table 2 ijerph-18-10039-t002:** Observed distribution of COVID-19 cases of ABO blood groups, incident rate, and relative risk comparing with blood donors, and an active general population from the Spanish Mediterranean zone as the reference population. Borriana COVID-19 cohort 2020.

Variables	ABO Blood Groups *n* = 483				*p*-Value
	O *N* (%)	A *N* (%)	B *N* (%)	AB *N* (%)	
Observed distribution in the cohort	200 (41.41)	220 (45.55)	44 (9.11)	19 (3.93)	
Blood donors ^1^	------------------------	------------------------	------------------------	------------------------	
Expected in reference population	229 (47.34)	203 (42.02)	36 (7.52)	15 (3.12)	0.271
COVID-19 Outbreak *n* = 1338					
Expected in reference population	633 (47.34)	562 (42.02)	101 (7.52)	42 (3.12)	
Incidence rate ×100	31.6	39.1	43.6	45.2	
Relative Risk 95% CI ^2^	1.00	1.24 (1.02–1.50)	1.37 (0.99–1.91)	1.43 (0.89–2.29)	0.008
Blood donors ^3^	-------------------------	-------------------------	-------------------------	--------------------------	
Expected in reference population	232 (48.03)	208 (42.99)	31 (6.43)	12 (2.55)	0.089
COVID-19 Outbreak *n* = 1338					
Expected in reference population	643 (48.03)	575 (42.99)	86 (6.43)	34 (2.55)	
Incidence rate ×100	31.1	38.3	51.2	55.9	
Relative Risk 95% CI ^2^	1.00	1.23 (1.02–1.49)	1.64 (1.19–2.28)	1.80 (1.12–2.88)	0.000
Active general population ^4^	-------------------------	-------------------------	-------------------------	------------------------	
Expected in reference popopulation	211 (43.76)	216 (43.76)	46 (9.73)	10 (2.0)	0.366
COVID-19 Outbreak *n* = 1338					
Expected reference population	585(43.76)	598 (43.76)	128 (9.73)	27 (2.0)	
Incidence rate ×100	34.2	36.8	34.4	70.4	
Relative Risk 95% CI	1.00	1.08 (0.89–1.30)	1.06 (0.72–1.39)	2.06 (1.29–3.30)	0.084

^1^ [13] Muñiz-Diaz, E. et al. *Blood Transfus.* **2021**, *19,* 54–63. ^2^ CI: confidence interval. ^3^ [29] Zalba-Marcos, S. et al. *Med. Clin. (Engl. Ed.)* **2020**, *155*, 340–343. ^4^ [30] Nogareda-Barbudo. *An. Med. Cir.* **1964,**
*44*, 115–123.

**Table 3 ijerph-18-10039-t003:** Symptoms in the acute COVID-19 illness in March 2020 and previous health status and health status in October 2020 by ABO blood groups. Borriana COVID-19 cohort 2020.

Variables	ABO Blood Groups *n* = 483				*p*-Value
	O *N* = 200 *N* (%)	A *N* = 220 *N* (%)	B *N* = 44 *N* (%)	AB *N* = 19 *N* (%)	
Symptomatic	178 (89)	193(87.7)	41(93.2)	17 (89.5)	0.804
Asymptomatic	22 (11)	27 (12.3)	3 (14.7)	2 (10.5)	
Medical consultation for acute illness	86 (43)	94 (42.7)	23 (52.3)	4 (21.1)	0.152
Hospitalization	3 (1.5)	4 (1.8)	2 (4.5)	0 (0)	0.542
Illness duration ^1^	11.7 ± 13.4	13.2 ± 22.5	13.7 ± 14.9	11.3 ± 10.8	0.911
Illness symptoms					
Cough ^2^	85 (42.5)	93 (42.3)	16 (36.4)	8 (42.1)	0.899
Runny nose	53 (26.5)	63(28.6)	10 (22.7)	8 (42.1)	0.436
Throat pain	56 (28)	66 (30)	16 (36.4)	7 (36.8)	0.618
Fever	98 (49)	94 (42.7)	30 (68.2)	9 (47.4)	0.020
Loss smell/taste ^3^	100 (50)	100(45.5)	31 (70.5)	7 (36.8)	0.016
Diarrhea	44 (22)	52 (23.6)	16 (36.4)	5 (26.3)	0.247
Vomits	7 (3.5)	13 (5.9)	5 (11.4)	1 (5.3)	0.158
Weakness	101 (50.5)	104 (47.3)	29 (65.9)	11 (57.9)	0.137
Headache	76 (38)	93 (42.3)	23(52.3)	7 (36.8)	0.379
Myalgia	91 (45.5)	96 (43.6)	25 (56.8)	12(63.2)	0.189
Dyspnea	8 (4)	6 (2.7)	1 (2.3)	2 (10.5)	0.282
Skin’s lesions ^4^	18 (9)	23 (10.5)	5 (11.4)	1 (5.3)	0.861
Habitual health status					
Poor	1 (0.01)	0 (0)	0 (0)	0 (0)	0.743
Fair	7 (3.5)	8 (3.6)	4 (9.1)	0 (0)	
Good	106 (53)	119 (54.1)	24 (54.5)	10 (52.6)	
Very good	85 (42.5)	89 (40.5)	16 (36.4)	9 (47.4)	
Health status October 2020					
Poor	0 (0)	1 (0.01)	0 (0)	0 (0)	0.595
Fair	18 (9)	16 (7.3)	5 (11.4)	0 (0)	
Good	139 (69.5)	144 (65.5)	28 (63.6)	12 (63.2)	
Very good	43 (21.5)	59 (26.8)	11 (25.0)	7 (36.8)	

^1^ Missing information 28 participants; ^2^ missing information 6 participants; ^3^ missing information 1 participant; ^4^ missing information 21 participants.

**Table 4 ijerph-18-10039-t004:** Complications and now reported symptoms post-COVID-19 and ABO blood group. Borriana COVID-19 cohort 2020.

Variables	ABO Blood Group				*p*-Value
	O (*N* = 200) *N* (%)	A (*N* = 220) *N* (%)	B (*N* = 44) *N* (%)	AB (*N* = 19) *N* (%)	
Complications	63 (31.5)	70 (31.8)	22 (50.0)	4 (21.1)	0.072
Recovery health ^1^	166 (83.0)	179 (81.4)	31 (70.5)	19 (100)	0.047
Health the same as before ^2^	173 (86.5)	178 (80.9)	32 (72.7)	19 (100)	0.025
Medical consultation ^3^	19 (9.6)	23 (10.6)	5 (11.9)	0 (0)	0.539
Complications‘ duration (mean ± standard deviation)	163.0 ± 47.0	160.3 ± 45.8	158.3 ± 44.6	150.0 ± 52.0	0.919
Reported symptoms October 2020					
Reported at least one symptom	101 (50.5)	119 (59.1)	27 (61.5)	11 (57.9)	0.586
Fatigue	32 (16.0)	40 (18.1)	13 (29.6)	2 (10.5)	0.187
Weakness	14 (7.0)	17 (7.7)	6 (13.6)	3 (15.8)	0.240
Dyspnea	13(6.5)	16 (7.3)	3 (6.8)	1 (5.3)	0.986
Thorax oppression	6 (3.0)	8 (3.6)	2 (4.2)	1 (5.3)	0.716
Cough	13 (6.5)	10 (4.6)	2 (4.6)	1 (5.3)	0.810
Fever	3 (1.5)	2 (0.9)	0 (0)	0 (0)	0.830
Throat pain	9 (4.5)	15 (6.8)	4 (9.1)	0 (0)	0.424
Runny nose	15 (7.5)	18 (8.2)	3 (6.8)	1 (5.3)	0.988
Loss smell/taste	30 (15)	35 (15.9)	11(35)	2 (10.5)	0.390
Nausea/vomits	4 (2.0)	2 (0.9)	1 (2.3)	0 (0)	0.561
Diarrhea	8 (4.0)	11 (5.0)	1 (2.3)	1 (5.3)	0.817
Alimentary intolerance	4 (2.0)	4 (1.8)	2 (4.6)	0 (0)	0.592
Abdominal pain	9 (4.5)	6 (7.3)	4 (9.1)	1 (5.3)	0.179
Muscle pain	21(10.5)	27 (12.3)	8 (18.2)	0 (0)	0.196
Headache	30 (15)	27 (12.3)	13 (29.6)	2 (10.5)	0.045
Hand/Foot pain	16 (8.0)	18 (8.2)	2 (4.6)	0 (0)	0.664
Dizziness	9 (4.5)	8 (3.6)	4 (9.1)	1(5.3)	0.348
Ringing ears	8 (4.0)	13 (5.9)	2 (4.6)	0 (0)	0.277
Disorder vision	6 (3.0)	8 (3.6)	4 (9.1)	0 (0)	0.183
Insomnia	25(12.5)	22 (10.0)	7 (15.9)	2 (10.5)	0.634
Night sweats	11 (5.5)	16 (7.3)	2 (4.6)	0 (0)	0.734
Depression	9 (4.5)	5 (2.3)	1 (2.3)	0 (0)	0.593
Restlessness	18 (9.0)	16 (7.2)	4 (9.1)	1 (5.3)	0.913
Difficulty concentration	10 (5.0)	7 (3.2)	3 (6.8)	0 (0)	0.544
Anxiety	18 (9.0)	21 (9.6)	4 (9.1)	1 (5.3)	0.991
Mental confusion	9 (4.5)	6 (2.7)	1 (2.3)	0 (0)	0.764
Difficulty articulating words	1 (0.5)	3 (1.4)	2 (4.6)	0 (0)	0.181
Difficulty to solve simple math operations	2 (1.0)	0	2 (4.6)	0 (0)	0.030
Skin lesions	11 (5.0)	8 (3.6)	6 (13.6)	0 (0)	0.059
Hair lost	48 (24.0)	51 (23.2)	15 (34.1)	5 (26.3)	0.472

^1^ Missing information 1 participant; ^2^ missing information 1 participant; ^3^ missing information 5 participants.

**Table 5 ijerph-18-10039-t005:** Adjusted incidence rate (aIR), adjusted relative risk (aRR), and 95% confidence interval (CI) by inverse probability weight regression of complications, and reported symptoms of ABO blood groups. Borriana COVID-19 cohort 2020.

Variables	aIR ^1^ (%)	aRR ^1^	95% CI	*p*-Value
Complications				
O	32.3	1.00		
A	32.6	1.01	0.77–1.33	0.950
B	54.1	1.68	1.24–2.27	0.001
AB	27.6	0.24	0.01–4.90	0.351
Recovery of illness				
O	83.1	1.00		
A	81.4	0.98	0.89–1.07	0.621
B	72.8	0.86	0.71–1.04	0.127
AB	100.0	1.20	1.13–1.28	0.000
Health the same as before				
O	86.4	1.00		
A	79.9	0.93	0.85–1.01	0.077
B	73.7	0.85	0.70–1.03	0.103
AB	100.0	1.16	1.10–1.22	0.000
Reported symptoms				
Fatigue				
O	16.2	1.00		
A	19.9	1.22	0.81–1.85	0.331
B	29.0	1.79	1.08–2.95	0.023
AB	2.6	0.16	0.01–5.23	0.305
Muscle pain				
O	9.8	1.00		
A	13.0	1.32	0.79–2.22	0.292
B	21.1	2.06	1.10–3.84	0.023
AB	0.0	NC ^2^		NC ^2^
Loss smell/taste				
O	14.8	1.00		
A	16.5	1.11	0.71–1.74	0.652
B	26.0	1.75	0.95–3.23	0.070
AB	15.1		0.01–9.28	0.524
Headache				
O	13.9	1.00		
A	12.9	0.93	0.57–1.50	0.751
B	36.4	2.61	1.58–4.31	0.000
AB	20.7	1.49	0.60–3.71	0.392
Skin’s lesions				
O	5.5	1.00		
A	3.4	0.71	0.29–1.72	0.448
B	14.1	2.46	0.98–6.20	0.056
AB	0	NC ^2^		NC ^2^
Disorder vision				
O	3.0	1.00		
A	3.8	1.29	0.46–3.61	0.624
B	12.8	4.26	1.33–13.60	0.014
AB	0	NC ^2^		NC ^2^
Brain fog ^3^				
O	12.3	1.00		
A	8.4	0.69	0.38–1.20	0.188
B	18.1	1.46	0.75–2.73	0.266
AB	5.9	0.48	0.13–1.71	0.257
Medical consultation acute illness				
O	42.5	1.00		
A	44.5	1.05	0.85–1.29	0.678
B	56.7	1.33	1.01–1.75	0.041
AB	31.9	0.75	0.44–1.28	0.294

^1^ Adjusted for age, sex, smoking, alcohol consumption, body mass index, physical exercise, person cough mass gathering events (MGE), contact COVID-19 case, family COVID-19 case, and attendance MGE ≥ 2 and over. ^2^ NC = not computable. ^3^ Brain fog = difficulty concentration + mental confusion + restlessness + difficulty articulating words + difficulty simple math operations.

## Data Availability

Data of this study can be consulted if the authors are requested. Dataset: borrianacohort.dta.

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
