# Peer review of "ABO Blood Groups and the Incidence of Complications in COVID-19 Patients: A Population-Based Prospective Cohort Study"

_ijerph, 2021, doi:10.3390/ijerph181910039_

Round 1

Reviewer 1 Report

This study is about the links between ABO blood groups and Covid-19. Although a lot has already been published on that topic, this work is original in that it focuses on the post-Covid symptoms on a cohort of patients contaminated during a festival that took place in Spain in March 2020. One of the main difficulties in this kind of study is to find a good reference group for ABO frequencies. I would say that the best in this situation would have been to use the SARS-Cov2 negative individuals who attended the same event, which is obviously practically difficult to do. Nonetheless, the authors have instead used three different reference populations from the neighboring regions, which should be fairly representative of the ABO frequencies in the cohort. However, the third group had been described in a paper from 1964, which seems rather old. I reckon the composition of the Spanish population has substantially changed in the last 60 years (by I would say “population of the Spanish Mediterranean zone” otherwise it is a bit confusing).  The ABO frequencies for this group are indeed somewhat different from the two groups of blood donors and I wonder if it really makes sense to use these data.

Regarding ABO and sensibility to SARS-CoV2 infection, I find the authors are too negative about their results. Although there is indeed no statistically significant difference in the overall distribution of ABO groups between the patients and the reference populations, the increased relative risk in the non-O vs O groups is statistically significant at least when comparing to the two first reference population (and close to significance with the third one, which as I said is not the most relevant one). This is in agreement with most of the published studies and I would thus mention this result in the abstract and discussion.

The main finding of the study is the higher frequency of post-Covid symptoms in the B group individuals. The authors mention the paper of Bloch et al who found higher titers of neutralizing antibodies in the B group individuals (but levels of the overall anti-SARS-CoV2 antibody response similar to the other groups). However, it seems to me that it should help them to clear the virus better and thus I would rather expect that they are less affected by post-Covid symptoms. Could the authors precise their views on this?

Minor points :

  • Could the authors indicate whether the same group of people attended the different events from the Falles festival or whether these were independent events? If the same group of individuals attended the different events, the situation could indeed be compared to the situation in the aircraft carrier described by Boudin et al.
  • Line 30: “…until October 2020…”
  • Line 96: replace “assistance” by “attendance. Suppress “and over”
  • Line 148: “…a decrease of Very Good…”
  • Line 149: “…there was no difference…”
  • Line 155 “March
  • Line 17: “…and lower for recovery and return health the same as before” (I think it would be clearer to write the items of the tables in italic or in quotation marks)
  • Line 193 “…could be found…”
  • Line 195: “…model of ABO incompatibility…”
  • Line 199: “…a few on the symptoms…”
  • Table 2: I think the table would be easier to read if there were lines to separate the Observed distribution in the cohort and each of the three groups of analysis.

Author Response

Attach.

Reviewer 2 Report

This study analyzed the association of ABO blood group with risks of COVID-19 symptoms and morbidity. 

The analysis has demonstrated that COVID-19 patients associated with the B blood group have higher risks of severe disease.

The study is interesting however I have the following comments that I hope will be useful for improving the quality of the manuscript.

1) The manuscript should analyze the mortality risk between ABO blood groups: is the B group associated with higher mortality for COVID-19 disease?

2)  Symptoms for Long COVID-19 (muscle pain, fatigue, and brain fog) should be also considered in the analysis.  

3) The manuscript should be reviewed for language and clarity, some examples are the following:

a) Line 69. "The aim of this study was to study the frequency..."

b) Line 74. "...during Mars 2020..."

c) Line 76. "...in relation to participants` ABO." Please add "blood group".

d) Line 145. "...with significant differences for fever and loss of smell/taste (p=0.020) and (p=0.016)."

e) Line 146-149. Please improve the sentence clarity.

f) Line 155. "period Mars-October 2020"

g) Line 199. "...few i the symptoms.."

h) Line 217. "In addition, Bloch and 217 co-authors [51] found that blood donors of group B had higher neutralizing antibody titers than the other ABO, suggesting a more intense cross-reactivity between the virus and B-group." Please explain more clearly how cross-reactivity would induce higher neutralizing antibody titers.

i) Line 220. "This could be associated with a higher frequency of more complications and reported symptoms in the B-group subject." Please explain how cross-reactivity between the B group and the virus would induce more severe disease.

j) Line 229. "The study had some potent points..." I would replace the word "potent" with "important".

k) Line 233. "The study presents limitations, complications...". Please replace the comma with colon punctuation.  

Author Response

attach.

Round 2

Reviewer 2 Report

The authors have revised the manuscript according to the reviewer's comments.